# Synthetic data enables human-grade microtubule analysis with foundation models for segmentation

Mario Koddenbrock[1☯*], Justus Westerhoff[2☯*], Dominik Fachet[3], Simone Reber[2,3], Felix A. Gers[2], Erik Rodner[1,4*]

**1** KI Werkstatt, Hochschule für Technik und Wirtschaft Berlin (HTW), Berlin, Germany, **2** Berliner Hochschule für Technik (BHT), Berlin, Germany, **3** Max Planck Institute for Infection Biology, Berlin, Germany, **4** Merantix Momentum GmbH, Berlin, Germany

☯ These authors contributed equally to this work.
* mario.koddenbrock@htw-berlin.de (MK); justus.westerhoff@bht-berlin.de (JW); erik.rodner@htw-berlin.de (ER)

## Abstract

Studying microtubules (MTs) and their mechanical properties is central to understanding intracellular transport, cell division, and drug action. While important, experts still need to spend many hours manually segmenting these filamentous structures. The suitability of state-of-the-art methods for this task cannot be systematically assessed, as large-scale labeled datasets are missing. We address this gap by introducing the synthetic dataset `SynthMT`, produced by tuning a novel image generation pipeline on real-world interference reflection microscopy (IRM) frames of *in vitro* reconstituted MTs without requiring human annotations. In our benchmark, we evaluate nine fully automated methods for MT analysis in both zero- and Hyperparameter Optimization (HPO)-based few-shot settings. Across both settings, classical algorithms and current foundation models still struggle to achieve the accuracy required for biological downstream analysis on *in vitro* MT IRM images that humans perceive as visually simple. However, a notable exception is the recently introduced SAM3 model. After HPO on only ten random `SynthMT` images, its text-prompted version SAM3Text achieves near-perfect and in some cases super-human performance on unseen, real data. This indicates that fully automated MT segmentation has become feasible when method configuration is effectively guided by synthetic data. To enable progress, we publicly release the dataset, the generation pipeline, and the evaluation framework.

## Author summary

Understanding the behavior of microtubules — stiff filaments inside cells — is essential for studying fundamental cell biological processes and for developing therapies for diseases such as cancer and neurodegenerative conditions. Yet,

**Data availability statement:** All data and code underlying the findings of this study are publicly available without restriction. The dataset can be accessed on Hugging Face at https://huggingface.co/datasets/HTW-KI-Werkstatt/SynthMT. The complete computational code is available on GitHub at https://github.com/ml-lab-htw/SynthMT. Interactive examples and additional resources are provided on the project page at https://datexis.github.io/SynthMT-project-page/.

**Funding:** Our work is funded by the Deutsche Forschungsgemeinschaft (DFG, German Research Foundation) Project-ID 528483508 - FIP 12. The Reber lab thanks the Max Planck Society for funding. The funders had no role in study design, data collection and analysis, decision to publish, or preparation of the manuscript.

**Competing interests:** The authors have declared that no competing interests exist.

analyzing microtubule images is slow and labor-intensive, as researchers must manually trace these thin, overlapping filaments, which can take hours and is prone to errors. We therefore asked whether current fully automated segmentation methods are ready to replace manual microtubule analysis, and how synthetic data can be used to rigorously evaluate and improve them. To address these questions, we created a synthetic dataset that mimics real microtubule images by capturing the appearance and variability of real microscopy. Importantly, this dataset can be generated without any manual annotations. Using this dataset, we evaluated a range of segmentation methods and found that most of them struggled to accurately identify filaments. However, we discovered that a recent foundation model, when guided by a simple text instruction and tuned on only a few synthetic images, can achieve near-perfect, human-level performance on previously unseen, real microtubule imaging data. Our work demonstrates that fully automated microtubule analysis is now possible and provides a reproducible framework that other researchers can use to evaluate and improve their methods. This opens the door to faster, more consistent studies of microtubules, ultimately accelerating discoveries in cell biology and therapeutic research.

## Introduction

Microtubules (MTs) are cytoskeletal filaments essential for cellular processes such as chromosome segregation, intracellular transport, and cell motility. They are formed by head-to-tail polymerization of $\alpha\beta$-tubulin heterodimers into protofilaments that laterally associate to form a hollow cylinder with a diameter of about 25 $nm$ [1]. Their plus ends exhibit *dynamic instability*, stochastically switching between growth and shrinkage, a behavior exquisitely sensitive to regulatory proteins and small molecules. This sensitivity makes MTs prime targets in drug discovery: widely used chemotherapeutics (e.g., taxanes [2], vinca alkaloids [3]) exploit destabilization, while emerging neuroprotective strategies aim to stabilize MT dynamics in neurodegenerative diseases [4]. To investigate how candidate compounds, cofactors, or post-translational modifications alter growth behavior, curvature, and length distributions, MTs are often reconstituted *in vitro* from stabilized nucleation seeds and visualized using microscopy techniques like total internal reflection fluorescence (TIRF) or interference reflection microscopy (IRM). A key step in such analyses is the accurate quantification of filament properties (count, length, curvature), which requires precise instance segmentation. However, manual annotation of MT images is labor-intensive and error-prone, creating a significant bottleneck in experimental and preclinical workflows [5–7].

Automated segmentation methods offer a promising solution, but their development and evaluation are hampered by a major challenge: the lack of large-scale, publicly available, and annotated datasets for *in vitro* MT images that resemble the typical noise of biological imaging data. This data scarcity has been repeatedly noted [8–10], yet no comprehensive benchmark exists.

While general-purpose segmentation models, including the SAM-based CellSAM [13], $\mu$SAM [14], and Cellpose-SAM [15], have shown impressive zero-shot performance on various microscopy datasets, they are primarily designed for round objects such as nuclei or whole cells. Their applicability to filamentous structures such as MTs remains largely unproven, and they often lack the accuracy necessary for reliable downstream biological analysis, especially when dealing with out-of-distribution data [16–19]. This raises a central question: **Are current fully automated methods ready to replace manual analysis for MT segmentation?** As sufficiently advanced methods often critically depend on their hyperparameters, we also ask to what extent synthetic data that mimics real microscopy images can enable few-shot learning through Hyperparameter Optimization (HPO), thereby improving generalization to real-world MT data.

This paper addresses these questions by introducing `SynthMT`, a synthetic benchmark designed to systematically evaluate the readiness of segmentation methods for automated MT analysis in zero- and few-shot settings (see Fig 1). Our main contributions and results are:

- **Synthetic data generation pipeline replicating real images.** Our publicly available pipeline generates realistic MT images along with corresponding ground-truth annotations. For tuning its internal parameters, it only requires real MT images (IRM or total internal reflection fluorescence (TIRF)) without the need for human annotations.

- **Synthetic benchmark dataset (`SynthMT`).** We apply this generation pipeline to real IRM images to construct and release the `SynthMT` dataset including instance masks for each MT. Its biological plausibility is assessed through a study with domain experts.

- **Comprehensive evaluation.** We conduct a reproducible benchmark evaluation of classical and state-of-the-art segmentation methods on `SynthMT`, including zero-shot and HPO-based few-shot adaptation experiments, establishing quantitative baselines.

- **Enabling fully automated segmentation.** We show that the newly released foundation model SAM3 achieves human-grade performance on real, unseen MT images when parameter-optimized only via synthetic images from `SynthMT`. Our synthetic data thus makes it possible to adapt this general-purpose model to the specific domain of *in vitro* MT

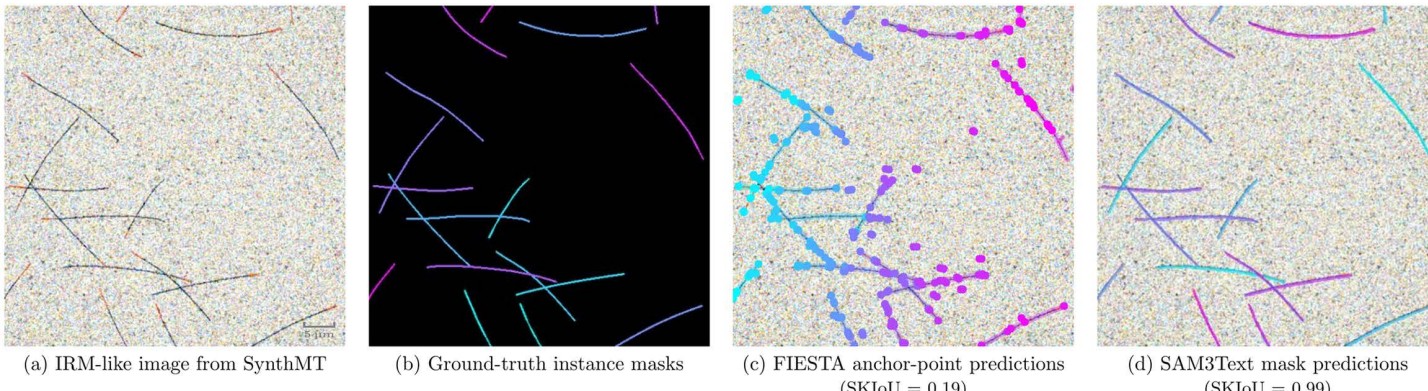

(a) IRM-like image from SynthMT  (b) Ground-truth instance masks  (c) FIESTA anchor-point predictions (SKIoU = 0.19)  (d) SAM3Text mask predictions (SKIoU = 0.99)

**Fig 1. The `SynthMT` instance segmentation benchmark evaluates methods on synthetic interference reflection microscopy (IRM)-like images containing microtubules (MTs). (a)** Synthetic image mimicking IRM of *in vitro* reconstituted MTs nucleated from fixed seeds (visualized in red), reproducing key mechanical and geometrical properties such as filament length and curvature. **(b)** Our pipeline generates accompanying ground-truth instance masks that enable quantitative evaluation. **(c)** The classical FIESTA [11] algorithm predicts anchor points for each instance (for visual clarity, only the first and last point of each instance are shown), which we connect through splines. The example demonstrates typical failure modes: filament fragmentation (single MTs split into multiple instances), incomplete segmentation, and artifacts at intersections. **(d)** SAM3 [12] guided by a simple text prompt ("*thin line*") produces precise, human-grade segmentation, accurately tracing intersecting MTs. This is supported by its high Skeleton Intersection over Union (SKIoU) for this specific image.

microscopy, establishing a practical route toward fully automated segmentation for large-scale experiments and subsequent geometric analyses.

To facilitate progress, we publicly release the generation pipeline, dataset, and evaluation framework at DATEXIS.github.io/SynthMT-project-page, providing the community with tools to benchmark existing methods and accelerate method development.

## Related work

### Synthetic datasets for microscopy

Synthetic data has recently been explored as a way to overcome the lack of annotated microscopy datasets. For example, AnyStar [20] employs a generative model to produce synthetic 3D training volumes of star-convex objects used to train StarDist [21] models, enabling zero-shot generalization across organisms without retraining. While powerful for spherical targets such as nuclei, the star-convex shape assumption makes this approach inapplicable to filamentous structures such as microtubules (MTs), where continuity and curvature priors are essential.

The concurrently released MicSim_FluoMT [22] dataset represents an important step toward synthetic benchmarks for *in vivo* MTs. It provides microscopy-like images together with ground-truth segmentation masks, generated using Cytosim [23] for filament dynamics and ConfocalGN [24] for imaging rendering (see Fig B in S1 Appendix for examples). Their evaluation focuses on architectures trained entirely from scratch, but does not address the performance of existing foundation models.

Building on the same physics-driven paradigm, simulators such as biobeam [25] and microsim [26] model image formation explicitly through point-spread function (PSF) convolution, photon statistics, and detector noise. However, as they focus on volumetric imaging and beam propagation, they primarily target volumetric 3D fluorescence modalities rather than the specialized yet common case of visualizing *in vitro* reconstituted MTs on a quasi-2D surface using interference reflection microscopy (IRM).

In contrast, purely data-driven approaches such as the diffusion-based generative model by Saguy et al. [27] produce visually convincing synthetic MT images. However, their method does not generate corresponding ground-truth labels, and hence the data cannot be used as a benchmark, and evaluation remains limited to qualitative inspection. Their focus lies on data augmentation for training Content-Aware Image Restoration (CARE) [28] models, rather than on providing a validated dataset for systematic comparison.

Several generative models synthesize medical or biological images by conditioning on segmentation masks or other geometric annotations. For example, SPADE [29] introduces spatially adaptive normalization to generate high-fidelity images from semantic masks, and BrainSPADE [30] applies this idea to biomedical MRI synthesis, showing that mask-to-image generation can support downstream segmentation. Similarly, ControlNet [31] guides diffusion models using additional input images, such as edges or segmentation maps, to specify where objects appear and how they are arranged, through an auxiliary control branch. SegGuidedDiff [32] extends this to anatomically informed medical image generation by concatenating segmentation masks at every denoising step. Despite architectural differences, all of these methods depend on manually annotated segmentation masks to guide the image generation process. In contrast, our approach does not need ground-truth annotations: it extracts statistical structure directly from MT images and produces synthetic images with segmentation masks through a simulation-based pipeline, removing the dependency on manual segmentation entirely.

DRIFT [7] recently introduced a recurrent approach for instance segmentation of filamentous objects, motivated by the way humans "pick and trace" filaments. To enable training and evaluation, they generate synthetic datasets of curved lines with varying widths and lengths. Although this constitutes a creative strategy to alleviate data scarcity, the resulting images are highly abstract — essentially binary line drawings — and deviate strongly from the visual statistics of real microscopy

data (see Fig B in S1 Appendix for examples). Their evaluation is centered on a custom recurrent model and comparisons to prior filament-tracing methods (SOAX [33], SIFNE [34,35], and [8]), but the data and benchmark are not designed to capture the variability in appearance of actual MT microscopy images, nor to evaluate the performance of foundation models for segmentation. In contrast, our image pipeline explicitly aligns with real images in terms of model features (by design) and the biological plausibility of the resulting `SynthMT` dataset is judged by domain experts. Additionally, we make this data openly accessible via Hugging Face for easy benchmarking or usage for training.

**Generalist segmentation models**

SAM-based [36] models for microscopy segmentation include CellSAM [13], $\mu$SAM [14], and Cellpose-SAM [15]. SAM has since been updated to SAM2 [37] and SAM3 [12], adding video segmentation and concept-specific text prompting. CellSAM employs a vision transformer to localize bounding boxes that are then passed to an adapted SAM module, removing the need for manual prompts. $\mu$SAM adds a new decoder trained on microscopy data. Cellpose-SAM combines a modified SAM encoder with the flow field prediction and gradient tracking of Cellpose [38]. Furthermore, StarDist [21,39] models are explicitly designed for segmenting star-shaped objects; although MTs are not star-convex, we include StarDist for completeness.

All of these models were benchmarked on a broad range of datasets [38,40–63], most of which target nuclei or whole-cell segmentation. A few include elongated or tubular structures (e.g., filamentous bacteria in Omnipose [42], DeepBacs [43], or mitochondria in EM datasets), but none involve cytoskeletal MTs, where geometric continuity and sub-pixel-width features are critical.

These models for general cellular and nuclear segmentation have reached a level of performance where further improvements are increasingly limited by annotation variability rather than model capacity. For example, Pachitariu et al. [15] report that Cellpose-SAM exceeds inter-annotator agreement on standard cell and nucleus benchmarks. In contrast, important challenges remain in settings like ours, where the objects of interest are elongated, filamentous structures, rather than compact objects like individual cells or nuclei.

**Microtubule-specific methods**

Several earlier works have proposed custom pipelines for MT segmentation, and in some cases also tracking, often involving their own (synthetic) datasets. These include FIESTA [11], SOAX [33], CARE [28], SIFNE [34], MTrack [6], DRIFT [7], the methods by Liu et al. [8,64,65] and Masoudi et al. [66], KnotResolver [67], the work by Laydi et al. [22], and TARDIS [10]. Classical approaches typically depend on high signal-to-noise ratio (SNR) data and require substantial manual input, keeping humans firmly in the loop. Machine-learning-based methods [7,8,10,22,64–66] attempt to address this limitation; however, their accessibility is often hindered by a reliance on proprietary MATLAB software rather than open-source Python environments. Furthermore, the specialized datasets used to develop these methods are not released to the public.

As a result, reproducibility and fair cross-method benchmarking remain limited. Crucially, none of these approaches have been evaluated against modern foundation models such as those built on SAM [36]. Within the MT community, a notable exception in terms of availability and generalization capabilities is TARDIS [10]. It combines CNN-based segmentation with graph-based instance segmentation and is released as open-source Python code that provides pretrained models, including one specialized for MTs in 2D total internal reflection fluorescence (TIRF) images. The authors of TARDIS report that compared to Amira [68], a commercial software suite for 3D visualization and analysis, annotation accuracy for MTs improves by 42% on 3D cryo-EM/EM datasets.

In this work, we choose FIESTA as a traditional baseline and add TARDIS as a pretrained domain-specific model. Our pipeline and evaluation framework are implemented in Python with open-source dependencies, enabling reproducible, cross-lab benchmarking while avoiding proprietary tools such as MATLAB (except for FIESTA).

We document all evaluated methods in section 1 in S1 Appendix, detailing implementation choices and practical requirements. Beyond benchmarking, `SynthMT` is designed as a practical resource for experimental biologists, providing accessible, validated tools for MT analysis.

## Mathematical framework for synthetic image generation

We formalize our image generation pipeline as a two-step stochastic process that generates synthetic microscopy images $I \sim P_\theta(I)$ conditioned on a parameter set $\theta$ as illustrated in Fig 2. These two steps are: (1) a purely geometric process imitating microtubule (MT) morphology, and (2) an imaging process that adapts these structures such that they resemble real images. This second step is largely inspired by the augmentation pipeline used in AnyStar [20]. Conceptually, it is also related to the degradation modeling strategy employed in CARE [28].

(1) Microtubule geometry.

Each MT is modeled as a polyline consisting of $n$ segments $\{W_k\}_{k=0}^{n-1}$. Together, these segments define a curve $f : [0, L] \to \mathbb{R}^2$. Each $W_k$ is specified by a segment length $\ell_k$ and a relative bend angle $\phi_k$, where $\ell_k$ is sampled from a Gaussian distribution $\mathcal{N}(\mu, \sigma^2)$ with $\mu, \sigma \in \theta$. Filament curvature is introduced through a stochastic evolution of the bend angles $\phi_k$, which evolve as

$$\phi_k = \phi_{k-1} + B_k \eta_k, \qquad \eta_k \sim \text{Gamma}(\alpha, \beta), \tag{1}$$

where $\alpha, \beta \in \theta$ are shape and scale parameters, and $B_k$ is a random variable with values in $\{\pm 1\}$ that determines whether the filament bends to the left or right at segment $k$. The sampling of $B_k$ is determined through a flip probability $p_{\text{flip}} \in \theta$ and the maximum number of flips $n_{\text{flips}} \in \theta$,

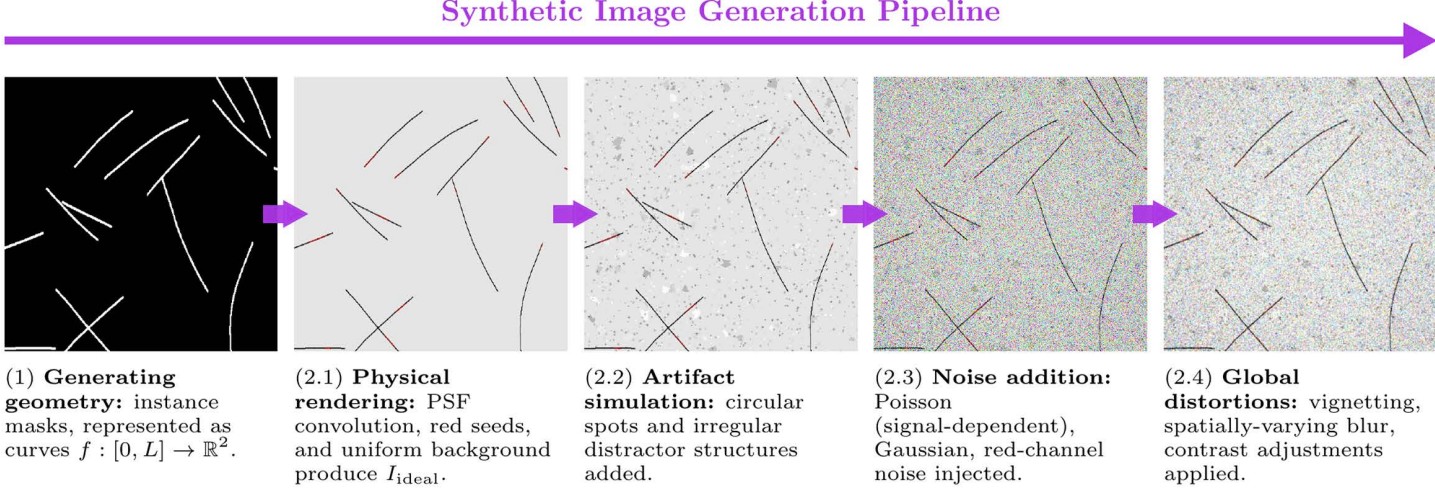

## Synthetic Image Generation Pipeline

(1) **Generating geometry:** instance masks, represented as curves $f : [0, L] \to \mathbb{R}^2$.

(2.1) **Physical rendering:** PSF convolution, red seeds, and uniform background produce $I_{\text{ideal}}$.

(2.2) **Artifact simulation:** circular spots and irregular distractor structures added.

(2.3) **Noise addition:** Poisson (signal-dependent), Gaussian, red-channel noise injected.

(2.4) **Global distortions:** vignetting, spatially-varying blur, contrast adjustments applied.

**Fig 2. Our synthetic data generation pipeline produces realistic microtubule (MT) images with corresponding instance segmentation masks conditioned on a parameter set $\theta$.** (1) Generating geometry creates instance masks from geometric parameters (count, length, curvature) using polylines. (2.1) Physical rendering applies point-spread function (PSF) convolution to replicate optical properties, and adds red seeds and uniform background. (2.2) Artifact simulation introduces realistic distractor features (circular spots, irregular structures). (2.3) Noise addition models signal-dependent (Poisson) and signal-independent (Gaussian) noise sources. (2.4) Global distortions apply spatially-varying effects (vignetting, blur, contrast variations) to match real microscopy conditions. This approach enables the generation of labeled data that closely approximates experimental interference reflection microscopy (IRM) images, when its set of generation parameters $\theta$ is tuned accordingly (as explained in section 4).

$$P(B_k \neq B_{k-1}) = p_{\text{flip}}, \quad \sum_{k=0}^{n-1} \left| B_k - B_{k-1} \right| \leq 2 \cdot n_{\text{flips}}.$$

(2)

This yields a correlated random walk with persistence length governed by $(\alpha, \beta)$. The overall filament length $L$ is drawn from a Gaussian distribution parametrized by $\theta$, and the initial seed orientation $\phi_0$ is sampled uniformly from $[0, 2\pi)$. Because segments are allowed to span only a few pixels, after step (2) they form smoothly curved filaments.

(2) Image rendering.

Given a curve $f$ as constructed before (i.e., a piecewise-linear MT skeleton), we compute a binary mask $M_S$ of shape $(h, w) \in \mathbb{N}^2$ that marks all pixels intersected by $f$. This mask is then convolved with the PSF $psf$, and scaled by a contrast factor $A \in \theta$ and background intensity $B \in \theta$,

$$I_{\text{ideal}}(x) := A\,(M_S * psf)(x) + B$$

(3)

for every pixel $x \in ([0, h) \times [0, w)) \cap \mathbb{N}^2$. Through convolution with the PSF, intensity spreads to neighboring pixels, which determines the filament's physical width in the rendered image.

Finally, noise and imaging artifacts are introduced via a stochastic operator $\nu_\theta$ that combines multiple perturbation processes. These include signal-independent noise (e.g., additive Gaussian noise), signal-dependent effects (e.g., Poisson shot noise and multiplicative speckle), channel-specific distortions, spatially correlated background variations, and structured artifacts such as vignetting, blur, and random distractor spots. All of these random variables instantiated by the noise process are collected in $\theta$ as well. This yields the final image $I$ through

$$I := \nu_\theta(I_{\text{ideal}}).$$

(4)

**Two-step stochastic process**

Together, the complete pipeline defines a probability distribution $P_\theta(I)$ over synthetic images, where $\theta$ encodes the distribution of the number of MTs, their length distribution and curvature statistics, and further imaging parameters. In principle, this framework also extends to a temporal process by defining a Markov chain $\{I_t\}$ (e.g., including MT dynamics such as growing, shrinking, pause with Gaussian growth/shrinkage increments at each time step). In the present work, however, we restrict to the *single-frame setting*, that is, single-sampling $I \sim P_\theta$ without simulating multi-frame temporal dynamics.

## Methods

We describe the creation of `SynthMT`, the first dataset of annotated, synthetic *in vitro* microtubule (MT) microscopy images that closely mimic the visual characteristics of real experimental data. Furthermore, we provide the protocol for assessing their perceptual realism using domain expert judgments and for evaluating fully automated segmentation methods on both `SynthMT` and a small labeled set of real images.

### Creating `SynthMT` and annotating real data

With the mathematical formulation of the image generation process in place (see section 3), we summarize the full pipeline to construct `SynthMT`, illustrated in Fig 3: For a given reference distribution $Q$ consisting of real, unlabeled images, the generator $P_\theta$ iteratively creates images $I \sim P_\theta(I)$ that are compared to samples from $Q$. For this comparison, we embed the images using DINOv2 [69], a pretrained state-of-the-art vision transformer, designed to generate rich visual

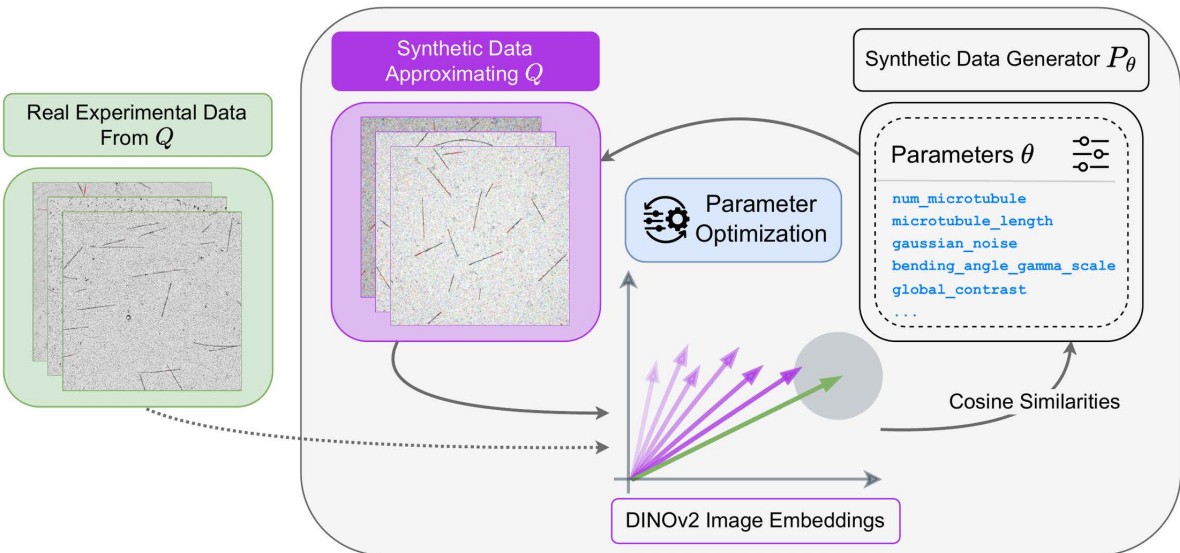

**Fig 3. Optimizing $\theta$ aligns synthetic image distributions with real, annotation-free microscopy data.** Real interference reflection microscopy (IRM) images (left) and synthetic images (center) are embedded using DINOv2. The parametric generator $P_\theta$ (right) creates images by sampling from distributions governing geometric properties (filament count, length, curvature) and imaging characteristics (PSF, noise, artifacts, contrast, distortions), all controlled by $\theta$. An optimization loop iteratively refines $\theta$ by maximizing cosine similarity between real and synthetic embeddings, ensuring that synthetic images match the statistical properties and visual characteristics of experimental data.

features. The pipeline parameters $\theta$ are then tuned to minimize the resulting embedding distance, without requiring ground-truth annotations.

## Real data

The real images for our empirical target distributions $Q_i$ were provided by two wet labs that routinely investigate reconstituted MTs. In total, we obtained 44 real IRM videos, each containing between 30 and 360 frames, spanning multiple experimental conditions and microscope setups, and capturing substantial diversity in MT geometry and visual properties such as contrast, noise, pixel intensity distributions, and background textures. Four representative examples of this variability are shown in Fig 4. More information about this real data can be found in section 3 in S1 Appendix. These raw videos were histogram-normalized and background-subtracted (as detailed at imagejdocu.list.lu/gui/process/subtract_background), with no additional processing applied.

From these videos, we take random $512 \times 512$ pixel crops. We manually discard crops containing significant imaging artifacts or distortions that would typically be excluded from a standard analysis, such as out-of-focus regions, all-black frames, or severe optical distortions, yielding a final set of 66 smaller videos. Each cropped video is treated as a distinct target domain $Q_i$, $i = 1, \ldots, 66$, from which we sample 10 random frames to serve as reference images for optimization. Importantly, for later method evaluations on unseen, real data, one of these frames from each crop is manually annotated by two independent human annotators using Label Studio [70]. In the analyses, we designate one annotator's labels as ground truth and treat the other annotator's labels as predictions, yielding an inter-annotator agreement score that quantifies the inherent variability of manual MT segmentation. This human baseline provides a reference point for assessing whether automated methods achieve human-level accuracy. Fig 4 shows four exemplary frames together with the annotations used as ground truth. Additionally, this subset of annotated real images is available on Hugging Face at huggingface.co/HTW-KI-Werkstatt/IRM-in-vitro-microtubules.

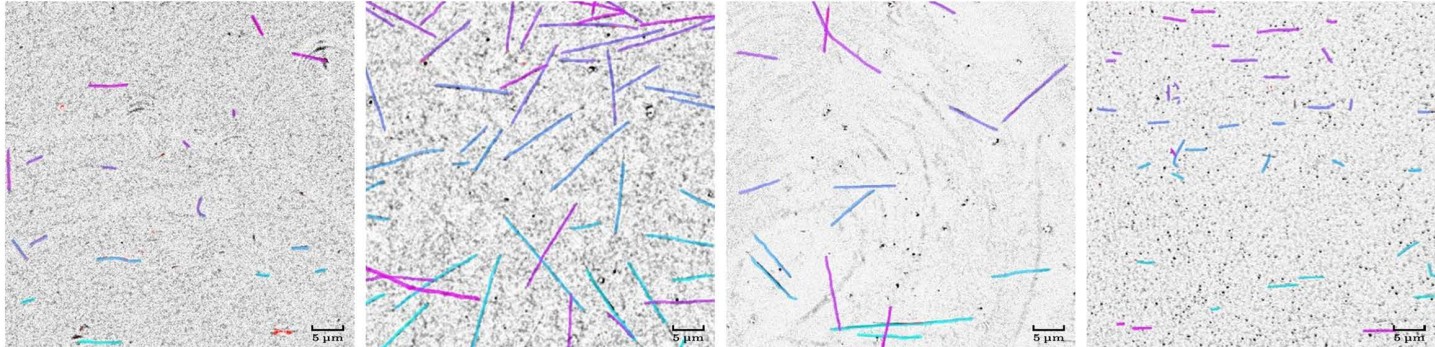

**Fig 4. Real IRM images with human ground-truth labels (for evaluation only).** Four exemplary frames from the 66 video crops of size $512 \times 512$ used to establish target distributions $\{Q_i\}_{i=1}^{66}$ (each containing 10 randomly sampled frames). Each image shows individual *in vitro* MTs growing from stabilized seeds under different experimental conditions, exhibiting natural variation in quantity, length, curvature, overlapping MTs, contrast, noise characteristics, filament density, and background properties. These target distributions act as references in the optimization process (Fig 3), where DINOv2 embeddings of $Q_i$ guide the synthetic data generation. Ground-truth labels from one annotator are overlaid to illustrate the filament structures present in the real data; they are used solely for later method evaluation and are not required for generating the synthetic dataset SynthMT.

## Optimizing $\theta$

To align statistical and visual properties of the generated synthetic images with those of the selected real microscopy data $\{Q_i\}_{i=1}^{66}$, we utilize the tree-structured Parzen estimator (TPE) [71] with 1000 iterations for each $i = 1, \ldots, 66$. TPE has repeatedly been shown to outperform alternative optimization algorithms [72,73] and has become the de-facto standard in recent Hyperparameter Optimization (HPO) frameworks, where it is commonly used as the default sampling strategy [74,75].

As a shared embedding space for comparing generated and real images, we choose the self-distillation method DINOv2 [69], which correlates well with human similarity judgments [76]. Moreover, as shown by Bolya et al. [77], the most informative embeddings for perceptual similarity are often obtained from intermediate rather than the final layers. This observation aligns with our findings: features from the fifth DINOv2 layer, denoted DINOv2$_5$, balance abstraction and sensitivity to the subtle, fine-grained image structures that define IRM MT data. Altogether, the optimization maximizes the cosine similarity

$$\text{sim}(I, J) := \text{sim}_{\cos}\big(\text{DINOv2}_5(I), \text{DINOv2}_5(J)\big)$$

(5)

between generated images $I \sim P_\theta(I)$ and real samples $J$ from $Q_i$ in the feature space. We aggregate the similarity between synthetic and real frames using the *maximum similarity* across the 10 references in $Q_i$ as the objective. This guarantees that the chosen parameters yield synthetic images that are plausible for at least one representative frame in $Q_i$. Finally, we retain the top $k = 10$ candidate solutions $\{\theta_i^j\}_{j=1}^{10}$ that yield the highest aggregated similarity. SynthMT is the product of drawing 10 images for each $1 \leq j \leq 10$ and $1 \leq i \leq 66$, yielding 6600 images of size $512 \times 512$, examples of which are shown in Fig A in S1 Appendix. The masks are stored as 3D TIFF images, where each slice is the mask assigned to a single instance. This preserves overlapping objects correctly and resembles the output of SAM (see section 1 in S1 Appendix).

SynthMT is publicly hosted on Hugging Face under a stable organization account at huggingface.co/HTW-KI-Werk-statt/SynthMT. The dataset card provides versioning, update logs, and contact information, and we will maintain SynthMT by releasing documented updates if errors are discovered or extensions are added. Users can contact the corresponding authors to report issues, request corrections, or raise takedown concerns.

## Assessing the perceptual realism of `SynthMT`

We conduct a study with $n = 6$ domain experts to evaluate the quality and realism of our synthetic microscopy images. To enable a direct comparison, we include 10 real images, 10 synthetic images from `SynthMT`, and 10 images from the only openly available synthetic dataset for *in vitro* MTs, DRIFT [7]. All 30 images are presented in randomized order and each image is shown exactly once to each participant. The study is administered as a web-based interface (see section 5 in S1 Appendix for details). Two of the six participants are co-authors of this paper; neither was involved in the data generation pipeline.

Experts rate each image on five aspects $\mathcal{D}$ using a 7-point Likert scale (1 = very poor, 7 = excellent): (a) Shape and general appearance of the MTs (structural fidelity), (b) Realism of the background, (c) Lighting, intensity, and blurring realism, (d) Noise pattern realism, and (e) Overall quality. The five aspects are motivated by quality evaluation criteria for microscopy imaging [78], while the use of aspect-specific Likert ratings follows recent generative evaluation studies emphasizing structured perceptual assessment of image realism [79,80]. We use this setup rather than pairwise forced-choice comparisons [81–84] to obtain aspect-specific judgments that also inform future pipeline improvements.

## Distributional comparison of ratings

Experts differ systematically in their use of Likert scales (e.g., lenient vs. strict raters). To remove these fixed rater effects, we z-normalize each expert's scores prior to aggregation. In particular, for expert $e = 1, \ldots, 6$ and aspect dimension $d \in \mathcal{D}$, we compute the within-expert mean $\mu_{e,d}$ and standard deviation $\sigma_{e,d}$ across all images, and transform raw ratings $r_{e,d}$ into normalized scores

$$r_{e,d}^{\text{norm}} := \frac{r_{e,d} - \mu_{e,d}}{\sigma_{e,d}}.$$

(6)

We visualize the normalized ratings for real and synthetic images using violin plots per dimension (Fig 5), displaying the full rating density, median, and interquartile range. Because normalization is done per expert, these visualizations capture differences in perceived quality rather than rater-specific scale usage.

## Evaluations on `SynthMT` and real data

As discussed in the related work (section 2), we focus on the classical algorithm FIESTA [11] as a baseline, and compare the foundation models SAM [36], SAM2 [37], and SAM3 [12], as well as the SAM-based methods $\mu$SAM [14], CellSAM [13], and Cellpose-SAM [15] (all listed chronologically by release date). StarDist [39] has an improper geometric bias for this task but is considered for completeness, and TARDIS [10] is added as the only MT-specific foundation model. The implementation and parameters of each method are detailed in section 1 in S1 Appendix. All methods are run in their "automated" mode without further input from our side. The only exceptions are SAM3Text, for which we use our manually specified default prompt "*thin line*", and FIESTA, where we set the *full width at half maximum* (FWHM) parameter to 3. We ignore other methods that cannot work in a fully automated fashion or are not publicly available (e.g., Masoudi et al. [66] or MTrack [6]). Initial tests with a StarDist model trained on AnyStar [20] showed complete failure on our 2D data. The authors later verified (via a public GitHub issue in the AnyStar repository) that the method is inherently ineffective in this setting.

We evaluate two categories of metrics: segmentation quality and downstream biological performance, as detailed below.

## Preprocessing

We disable method-specific preprocessing steps and instead gather all of those in our own preprocessing function. For details, see section 1 and 2 in S1 Appendix. This ensures that all methods share the same preprocessing capabilities during hyperparameter tuning (see HPO below).

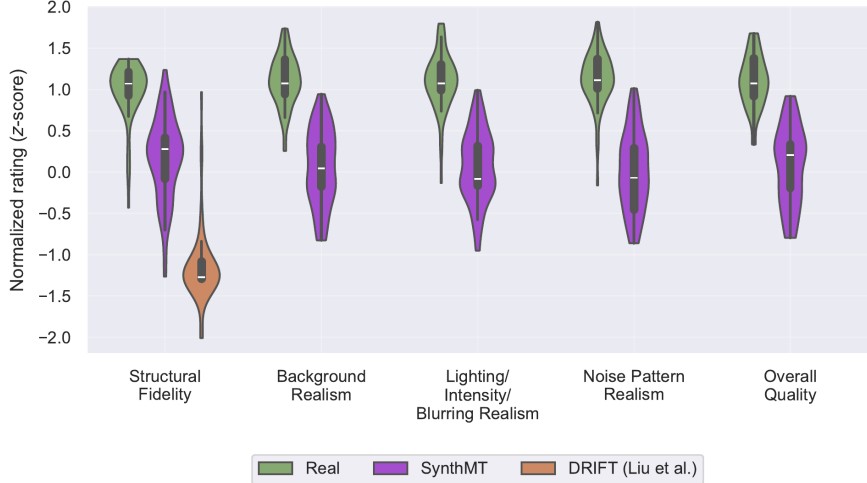

**Fig 5. Domain experts confirm perceptual quality of `SynthMT` images.** Violin plots show *z*-normalized ratings of *n* = 6 domain experts across five quality dimensions for real IRM images and synthetic images from `SynthMT` and DRIFT [7] (10 images each, 30 in total). Each violin displays the full distribution, median (white line) and interquartile range (thick bar). Ratings were collected on a 7-point Likert scale. DRIFT permits evaluation only of structural fidelity due to its black-and-white outputs (see exemplary images in Fig B in S1 Appendix). `SynthMT` images score higher than DRIFT on this dimension, indicating that parameter-optimized synthesis yields structures that more closely resemble real microscopy data. A measurable gap to real images persists across all dimensions. Nevertheless, experts rate the backgrounds, lighting, and noise patterns of `SynthMT` as internally coherent and plausibly aligned with real IRM, in contrast to DRIFT's limited realism.

## Method outputs

The methods differ in output format, which becomes crucial in postprocessing and metric computation. In particular, SAM, SAM2, and SAM3 return lists of $512 \times 512$ boolean masks, one per instance, which resembles the format of our labels for `SynthMT` and the real images we annotated. StarDist, $\mu$SAM, CellSAM, and Cellpose-SAM produce $512 \times 512$ integer arrays, where 0 denotes background and each positive integer corresponds to a unique instance. We convert these into our format by splitting the integer-labeled arrays into separate boolean masks, one per non-zero label. FIESTA and TARDIS natively return lists of anchor points $(p_k)_k$, one per instance, rather than pixel masks. While TARDIS can optionally produce binary semantic masks, we chose to use its anchor-point output for consistency with FIESTA and to avoid dependency on its internal drawing parameters (such as spline size and pixel size). Whenever we report instance lengths, for mask-based outputs $M$ our proxy is the skeleton pixel count $L = |S(M)|$, where $S(M)$ is the thinned, one-pixel-wide binary skeleton. For anchor points $(p_k)_k$, we sum Euclidean distances, $L = \sum_k \|p_{k+1} - p_k\|_2$.

## Postprocessing

To ensure comparability across methods, we also disable all method-specific postprocessing routines and apply a unified postprocessing step implemented by us as follows. Predicted instances are filtered by length and area (for anchor-point methods, only length is used), requiring them to fall within dataset-specific minimum and maximum values. In our benchmark, these thresholds are computed automatically from the ground-truth labels of `SynthMT` resp. the real data. In practical applications, however, domain experts typically know the plausible range of filament lengths and areas for their particular imaging setup, making such filtering a natural and interpretable step. By removing implausible tiny fragments and related artifacts, this unified postprocessing ensures more stable and meaningful comparisons across segmentation methods.

## Segmentation metrics

Accurately evaluating segmentation of thin, elongated MTs is non-trivial: classical pixel-overlap metrics such as Intersection over Union (IoU) over-penalize small transversal misalignments while under-representing errors in biologically relevant properties like filament length, continuity, or curvature. For MTs that are only a few pixels wide, a slight lateral shift or minor width disagreement can disproportionately reduce IoU, even if the traced centerline (and thus the inferred length) is essentially correct. To emphasize geometry rather than raw area agreement, we adopt the Skeleton Intersection over Union (SKIoU) [64] metric.

For anchor-point outputs (i.e., outputs from FIESTA and TARDIS), we compute a corresponding instance mask by fitting a spline using SciPy's splprep function with smoothing condition $s = 0$ through these points. We then represent this spline as a binary mask $M$ to match the format of the other methods.

Given predicted and ground-truth instance masks $M_\text{pred}$ and $M_\text{gt}$, and again denoting $S(M)$ as the skeleton of a mask $M$, their SKIoU is defined as

$$\text{SKIoU} := \frac{2 \left| S\left(M_\text{pred} \cap M_\text{gt}\right)\right|}{\left|S\left(M_\text{pred}\right)\right| + \left|S\left(M_\text{gt}\right)\right|}.$$

(7)

This provides a symmetric, length-normalized overlap over skeleton pixels: it attains 1 only when centerlines coincide, and it drops sharply when filaments are missing (under-segmentation), spuriously split or merged (topology errors), or substantially curved differently (shape distortion). Notably, the numerator first computes the mask intersection $M_\text{pred} \cap M_\text{gt}$ *before* skeletonization. As a result, small lateral shifts or minor thickness discrepancies still yield a substantial overlapping region whose skeleton preserves the shared filament topology, leading to robust SKIoU values.

For evaluation, we compute several common object detection and instance segmentation metrics based on the SKIoU values. We report the mean SKIoU of an image, calculated as follows. For each ground-truth instance, we find the best-matching prediction and record its SKIoU. If no matching prediction is found (a false negative), the score for that ground-truth instance is 0. The mean SKIoU value thus reflects segmentation quality while also penalizing for missed objects. While informative, this metric does not penalize false positives.

To provide a more complete picture, we also report Average Precision (AP) and F1 scores. AP is calculated as the mean of AP values at SKIoU thresholds from 0.5 to 0.95 in steps of 0.05, summarizing detection and segmentation accuracy. The F1 scores at specific SKIoU thresholds of 0.5 (F1@0.50) and 0.75 (F1@0.75) represent the harmonic mean of precision and recall. Comparing F1@0.50 and F1@0.75 allows us to gauge the consistency of a method's predictions; a small drop-off between these values indicates that the method maintains high localization accuracy even at stricter evaluation criteria.

## Downstream biological metrics

Segmentation scores alone do not answer whether a method performs well at recovering biologically relevant filament statistics. To accommodate that, we report downstream biological metrics derived from instance geometries such as **counting** the number of predicted instances and gathering their computed **lengths** (as explained above). Furthermore, we collect the average **curvature** of each predicted instance as follows: Anchor-point outputs are converted into masks by connecting the points via straight lines. Then, given any instance mask (directly from the method or converted from anchor points), we order skeletonized pixels by greedy nearest-neighbor and fit a parametric spline $(x(u), y(u))$ with smoothing condition $s = 0$ to preserve geometry. Finally, we compute the average curvature of this instance as the mean of the curvatures $\kappa(u)$, where

$$\kappa(u) := \frac{\left|x'(u)y''(u) - y'(u)x''(u)\right|}{\left(x'(u)^2 + y'(u)^2\right)^{3/2}}.$$

(8)

With these statistics at hand, we report mean and standard deviation across all images for counts, lengths, and curvatures. While this offers an intuitive understanding of systematic biases in the predictions relative to the ground-truth values, comparing distributions solely by their means and standard deviations is not sufficient. We therefore also compare the predicted and ground-truth distributions of length and curvature using normalized histograms with shared linear bins. Their dissimilarity is quantified through the Kullback–Leibler divergence (KL divergence) [85]

$$\mathrm{KL}(P \parallel Q) := \sum_i P_i \log \left( \frac{P_i}{Q_i} \right),$$

(9)

where $P$ and $Q$ denote the predicted and ground-truth discrete distributions, respectively (lower is better).

### Computational efficiency

We report computational throughput measured as processed images per second ("Img/s." in Table 1). All results are obtained under sequential, unbatched inference to reflect per-sample efficiency. Experiments are conducted on a single NVIDIA A100 GPU; FIESTA is evaluated separately on a MacBook M3 Pro.

### Zero-shot setting

We evaluate all methods with their default parameters (as detailed in section 1 in S1 Appendix) on `SynthMT` and the unseen, real data.

### HPO-based few-shot setting

Furthermore, we automatically adapt both method and preprocessing hyperparameters using only 10 random samples from `SynthMT`. We conduct this HPO via the tree-structured Parzen estimator [71] with 1000 trials to maximize mean SKIoU. We report the resulting optimal configurations in section 1 and 2 in S1 Appendix, and discuss optimization trajectories and parameter importance in section 8 in S1 Appendix. In section 7 in S1 Appendix, we further analyze the effect of increasing the number of HPO samples beyond 10 to assess potential saturation in performance gains. Additionally, we perform HPO directly on the binary DRIFT dataset [7] and on our human-annotated, real images.

## Results

The resulting `SynthMT` dataset, as described in section 4, contains 6600 synthetic images with a varying number and shape of microtubules (MTs), and diverse backgrounds and noise. They are accompanied with segmentation masks for every MT present in each image. Moreover, two independent human annotators created instance segmentations for 66 real interference reflection microscopy (IRM) images.

### Perceptual realism of `SynthMT`

The images in `SynthMT` were constructed to match real IRM-like data in the fifth layer representation of the DINOv2 model (see Eq (5)). For adoption of this dataset within the community, perceptual adequacy to humans matters as well. Here, we present the results of the expert study outlined in section 4.

   While a gap relative to real images remains, the distributions in Fig 5 nevertheless demonstrate that `SynthMT` occupies an intermediate but substantive region between DRIFT [7] and real data: `SynthMT` is geometrically closer to the real domain, perceptually coherent across background, lighting, and noise, and stable across experts. This establishes `SynthMT` as the most realistic synthetic alternative currently available.

**Table 1. Results on `SynthMT` signal unprecedented segmentation performance of the new SAM3 model.**

| Model | Img/s ↑ | SKIoU ↑ | AP ↑ | F1 ↑ @.50 | F1 ↑ @.75 | Count n/img | Length $\mu m$/MT | Length KL ↓ | Curvature $\mu m^{-1}$/MT | Curvature KL ↓ |
|---|---|---|---|---|---|---|---|---|---|---|
| *Traditional Baselines* | | | | | | | | | | |
| FIESTA [11] | 0.21 | 0.12 | 0.12 | 0.22 | 0.18 | 99.63 ± 56.0 | 1.92 ± 4.3 | 5.03 | 6.74 ± 12.6 | 0.997 |
| + HPO | 0.16 | 0.24 | 0.25 | 0.39 | 0.30 | 38.80 ± 17.1 | 3.64 ± 5.9 | 3.74 | 6.90 ± 10.2 | 0.706 |
| | | | | | | | | | | |
| StarDist [39] | 4.93 | 0.01 | 0.04 | 0.00 | 0.00 | 1.85 ± 1.5 | 10.14 ± 2.9 | 0.69 | 7.67 ± 2.1 | 0.196 |
| + HPO | **8.64** | 0.31 | 0.34 | 0.36 | 0.15 | 20.36 ± 11.0 | 2.65 ± 1.4 | 1.95 | 7.79 ± 7.8 | 0.183 |
| *Pretrained Domain-Specific Models* | | | | | | | | | | |
| TARDIS [10] | 0.32 | 0.45 | 0.48 | 0.59 | 0.45 | 17.15 ± 4.9 | 7.71 ± 6.5 | 0.56 | 8.28 ± 3.8 | **0.019** |
| + HPO | 0.34 | 0.48 | 0.55 | 0.57 | 0.41 | 12.33 ± 5.1 | 8.35 ± 6.3 | 0.41 | 8.50 ± 3.6 | 0.031 |
| *SAM-based Models* | | | | | | | | | | |
| $\mu$SAM [14] | **7.44** | 0.02 | 0.13 | 0.03 | 0.02 | 0.50 ± 1.0 | 6.03 ± 5.3 | 0.88 | 8.16 ± 8.1 | 0.130 |
| + HPO | 6.56 | 0.66 | 0.73 | 0.74 | 0.64 | 14.58 ± 22.9 | 7.22 ± 6.3 | 1.24 | 8.68 ± 10.5 | 0.132 |
| CellSAM [13] | 2.63 | 0.56 | 0.79 | 0.66 | 0.58 | 7.95 ± 4.6 | 9.57 ± 5.5 | 0.19 | 8.12 ± 4.1 | 0.021 |
| + HPO | 2.50 | 0.59 | 0.69 | 0.68 | 0.59 | **11.60 ± 6.7** | 9.33 ± 5.7 | 0.21 | 7.97 ± 4.5 | 0.031 |
| Cellpose-SAM [15] | 2.62 | 0.27 | 0.68 | 0.37 | 0.36 | 2.96 ± 3.2 | **12.54 ± 5.5** | 0.12 | **8.45 ± 4.2** | **0.019** |
| + HPO | 2.59 | 0.65 | 0.75 | 0.76 | 0.65 | 10.66 ± 6.0 | 10.39 ± 6.1 | 0.12 | **8.35 ± 4.9** | **0.012** |
| *Foundation Models* | | | | | | | | | | |
| SAM [36] | 0.72 | 0.37 | 0.46 | 0.45 | 0.44 | 20.46 ± 17.3 | 3.91 ± 5.9 | 3.90 | 9.09 ± 22.5 | 0.700 |
| + HPO | 0.52 | 0.16 | 0.16 | 0.25 | 0.24 | 74.90 ± 37.8 | 2.01 ± 4.7 | 5.45 | 7.52 ± 18.4 | 0.912 |
| SAM2 [37] | 0.96 | 0.01 | 0.05 | 0.01 | 0.01 | 0.06 ± 0.3 | 3.65 ± 2.8 | 1.62 | 6.90 ± 4.1 | 0.269 |
| + HPO | 0.62 | 0.66 | 0.73 | 0.74 | 0.72 | 11.67 ± 5.3 | 11.89 ± 6.8 | 0.04 | 8.18 ± 4.1 | 0.014 |
| SAM3 [12] | 1.00 | 0.00 | 0.00 | 0.00 | 0.00 | 0.01 ± 0.1 | 3.02 ± 1.1 | 2.77 | 9.50 ± 3.9 | 0.489 |
| + HPO | 0.78 | 0.54 | 0.58 | 0.64 | 0.64 | 23.69 ± 26.8 | 6.64 ± 8.1 | 2.90 | 8.29 ± 13.6 | 0.364 |
| SAM3Text [12] | 4.26 | **0.85** | **0.86** | **0.91** | **0.90** | **13.03 ± 6.5** | 11.74 ± 7.1 | **0.07** | 7.54 ± 3.3 | 0.063 |
| + HPO | 3.97 | **0.93** | **0.95** | **0.97** | **0.96** | 11.73 ± 5.6 | **13.19 ± 6.7** | **0.02** | 7.56 ± 3.0 | 0.069 |
| Ground Truth | – | 1 | 1 | 1 | 1 | 11.28 ± 5.1 | 13.35 ± 6.7 | 0 | 8.42 ± 3.9 | 0 |

For each method, we report segmentation metrics including skiou [64], Average Precision (AP) (mean over SKIoU thresholds), and F1 of SKIoU at 0.50 and 0.75. In addition, we report biological downstream metrics derived from counts per image, and length and curvature measurements per MT. These values are reported as mean ± standard deviation across all images and can be directly compared to the ground-truth statistics (last row). To capture distributional differences in greater detail, we report the Kullback–Leibler divergence (KL divergence) between the predicted and ground-truth distributions of length and curvature. All methods are evaluated zero-shot using their default parameters (white). Rows marked with "+ HPO" (gray) show performance after Hyperparameter Optimization (HPO) maximizing SKIoU on 10 random, synthetic `SynthMT` images, illustrating few-shot adaptation potential. SAM-family models are run in Automatic Instance Segmentation (AIS) mode; SAM3Text uses SAM3's text prompt mode (default prompt: "*thin line*"). For each column, the best-performing default and tuned methods are highlighted in **bold**, and ↑ (↓) signals higher (lower) is better.

In the following sections, we show that this perceptual adequacy has functional consequences too: methods' strengths on `SynthMT` transfer to unseen, real IRM data, and tuning via Hyperparameter Optimization (HPO) on only 10 images from `SynthMT` further improves performance on both synthetic and real data for many methods.

## Benchmarking `SynthMT` and real data

Beyond our generation pipeline used to produce `SynthMT`, Table 1 depicts another main contribution of this work: benchmark results for all nine introduced methods across the 6600 synthetic images contained in `SynthMT`. Additionally, we

present results on unseen, real data in Table 2, and a qualitative analysis in Fig 8. The benchmark is fully reproducible with the code available at github.com/ml-lab-htw/SynthMT.

**From FIESTA to SAM3Text**

From Table 1 we find that all deep learning-based methods perform substantially better than the traditional, deep learning-free baseline FIESTA [11] which is still state-of-the-art in many labs. As one of the fastest methods, SAM3Text with tuned hyperparameters emerges as the candidate of choice for almost all tasks. Notably, on `SynthMT` it is the only evaluated method that achieves sub-micrometer accuracy in MT length estimation. Fig 6 visualizes the close alignment between its predicted and ground-truth distributions of MT lengths and curvatures across scales. The only task where SAM3Text slightly lags behind is curvature estimation.

**Improvements through HPO**

In the few-shot (HPO) setting, we optimized method hyperparameters by maximizing SKIoU using only 10 random images from `SynthMT`. We find that this often greatly improves segmentation, except for TARDIS and CellSAM, which seem rather agnostic to parameters, and SAM, which shows a substantial drop in performance, indicating overfitting. Further analysis in section 7 in S1 Appendix shows that, for SAM3Text, increasing the number of synthetic images beyond $N = 10$ yields only marginal gains. Moreover, tuning on synthetic `SynthMT` images achieves parity with tuning on human-annotated, real images from the test set, confirming that the synthetic data provides a competitive, annotation-free adaptation resource. In contrast, tuning on simplified binary structures from DRIFT [7] leads to a substantial performance degradation, underscoring the importance of biologically realistic image formation.

**Localization robustness**

SAM3Text not only achieves the strongest segmentation scores overall, but also stands out for its internal consistency across complementary metrics. Its simultaneously high AP, SKIoU, F1@0.50, and F1@0.75 scores indicate that detected MTs are both reliably found and accurately localized, with little degradation when stricter overlap requirements are imposed. In particular, the stable performance from F1@0.50 to F1@0.75 suggests that SAM3Text predictions are geometrically precise. Other models from the SAM family exhibit a similar robustness pattern, albeit at a lower absolute performance level. In contrast, StarDist and TARDIS show a pronounced drop at F1@0.75, revealing inconsistencies between detection and precise alignment. A different failure mode is observed for CellSAM and Cellpose-SAM, which achieve comparatively high AP values while lagging behind in SKIoU. This discrepancy indicates that, although many MTs are detected, their spatial alignment is often imprecise, resulting in poorly localized segmentations. Such behavior is consistent with a higher incidence of missed or fragmented filaments in the later qualitative inspections.

**Segmentation metrics as proxies for biological downstream performance**

Overall, improvements in segmentation metrics translate into better performance on biological downstream tasks. However, this relationship breaks down for methods whose segmentation quality is fundamentally insufficient. For methods with near-zero segmentation scores, such as StarDist, $\mu$SAM, SAM2, and SAM3, even large gains in segmentation metrics achieved through HPO do not lead to meaningful improvements in biological performance and in some cases even degrade it. Apart from that, $\mu$SAM shows that poor performance on segmentation metrics is not immediately an indicator of bad downstream results. We hypothesize that this is because the default versions of SAM2, SAM3, Cellpose-SAM, and $\mu$SAM did not generate any predictions on up to 60% of the 6600 images, while their HPO versions did for almost all. Hence, $\mu$SAM identified only a small fraction of all MTs, yet statistically reproduced the correct distributions of MT geometries. We refer to Fig E in S1 Appendix and Fig F in S1 Appendix for these distributions for all methods, illustrating

**Table 2. Hyperparameter optimization on synthetic `SynthMT` images improves SAM3Text to human-grade performance on unseen, real IRM data.**

| | Biological Downstream Metrics | | | | |
| --- | --- | --- | --- | --- | --- |
| | Count | Length ↓ | | Curvature | |
| **Model** | n/img | $\mu m$/MT | KL ↓ | $\mu m^{-1}$/MT | KL ↓ |
| *Traditional Baselines* | | | | | |
| FIESTA [11] | 93.05 ± 77.4 | 1.15 ± 1.4 | 1.39 | **6.80 ± 11.8** | 0.27 |
| + HPO | 27.09 ± 28.0 | 1.87 ± 2.2 | 0.97 | **6.70 ± 11.1** | 0.28 |
| StarDist [39] | 0.68 ± 0.9 | 8.56 ± 2.4 | 1.46 | 7.66 ± 2.1 | 0.54 |
| + HPO | 33.76 ± 13.1 | 1.68 ± 1.1 | 0.61 | 8.19 ± 12.9 | 0.35 |
| *Pretrained Domain-Specific Models* | | | | | |
| TARDIS [10] | 10.20 ± 8.2 | 4.78 ± 3.5 | 0.32 | **7.42 ± 4.9** | **0.10** |
| + HPO | 8.85 ± 8.2 | 5.85 ± 4.4 | 0.41 | 7.71 ± 5.5 | 0.12 |
| *SAM-based Models* | | | | | |
| $\mu$SAM [14] | 1.27 ± 1.6 | 4.63 ± 3.4 | 0.29 | 6.64 ± 4.0 | 0.26 |
| + HPO | 67.26 ± 117.9 | 1.72 ± 2.3 | 1.10 | 7.71 ± 14.0 | 0.29 |
| CellSAM [13] | 21.29 ± 14.3 | 3.88 ± 2.7 | **0.07** | **7.08 ± 5.9** | **0.09** |
| + HPO | 16.73 ± 16.3 | **4.34 ± 3.0** | **0.08** | 8.25 ± 10.1 | 0.14 |
| Cellpose-SAM [15] | 12.27 ± 16.4 | 3.57 ± 2.7 | **0.05** | **7.03 ± 6.5** | 0.18 |
| + HPO | 13.00 ± 18.8 | 4.05 ± 3.3 | **0.07** | 7.92 ± 9.4 | 0.09 |
| *Foundation Models* | | | | | |
| SAM [36] | 28.55 ± 12.6 | 2.24 ± 2.6 | 0.79 | **6.80 ± 10.0** | 0.16 |
| + HPO | 108.26 ± 29.6 | 1.43 ± 2.1 | 1.39 | **6.62 ± 12.2** | 0.15 |
| SAM2 [37] | 2.73 ± 2.6 | 2.39 ± 1.6 | 0.37 | **7.43 ± 7.8** | 0.23 |
| + HPO | 48.79 ± 16.9 | 3.07 ± 3.3 | 0.44 | 7.74 ± 11.5 | 0.15 |
| SAM3 [12] | 0.03 ± 0.2 | 1.83 ± 0.4 | 1.45 | 8.29 ± 2.1 | 0.94 |
| + HPO | 16.59 ± 11.0 | **4.28 ± 3.5** | 0.10 | 7.71 ± 8.6 | 0.07 |
| SAM3Text [12] | 14.98 ± 15.4 | 5.41 ± 3.9 | 0.21 | **7.17 ± 5.3** | 0.25 |
| + HPO | **25.08 ± 16.9** | 4.69 ± 3.5 | **0.09** | **6.62 ± 4.9** | 0.14 |
| Inter-annotator | 25.29 ± 18.6 | 4.56 ± 3.4 | 0.09 | 7.43 ± 7.3 | 0.11 |
| Ground Truth | 23.91 ± 18.1 | 4.39 ± 3.6 | 0 | 6.91 ± 6.6 | 0 |

We evaluate all methods on unseen, real IRM images with biological downstream metrics (count, and length and curvature distributions; analogous to Table 1). Rows without "+ HPO" (white) report zero-shot performance with default parameters; rows marked "+ HPO" (gray) show performance after HPO on 10 synthetic `SynthMT` images, thereby evaluating the transferability of synthetic-data-guided optimization to unseen real IRM data. Additionally, we report an inter-annotator baseline by treating the annotations of a second, independent human annotator as predictions, quantifying inherent variability in manual segmentation. SAM3Text + HPO is the only method to reach human-grade performance, signalling the capability of fully automated MT analysis on real IRM data. Models that are on par with or surpass the human inter-annotator baseline are listed in **bold**.

the curvature distribution match of $\mu$SAM, even with poor predictions for count and length. Still, relying on downstream biological measurements when segmentation is weak comes at a risk, as they may be computed on noisy or incorrectly identified structures.

## Methods made specifically for microscopy are strong on this task

Although SAM3Text consistently performs well, it is important to note that Cellpose-SAM, as a SAM-based model specifically adapted for microscopy, also demonstrates strong performance on downstream metrics; even in its default setting (where it does not find any instances in 27% of the images). While TARDIS, $\mu$SAM, and CellSAM lag behind (in line

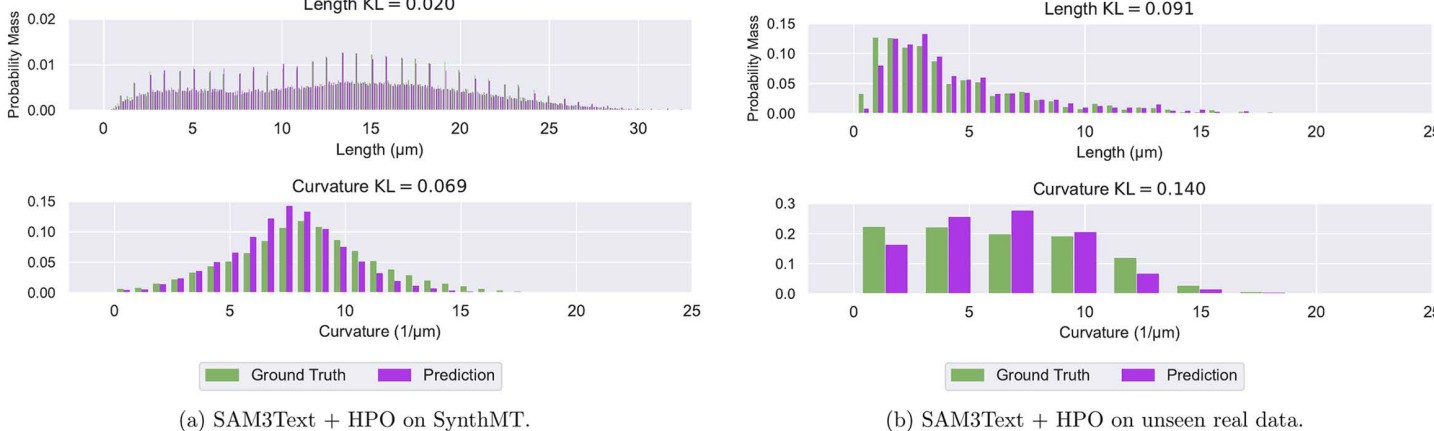

(a) SAM3Text + HPO on SynthMT.

(b) SAM3Text + HPO on unseen real data.

**Fig 6. SAM3Text+HPO closely matches ground-truth MT length and curvature distributions across scales and datasets.** Normalized histograms compare predicted and ground-truth distributions for MT length (top row) and curvature (bottom row) on **(a)** SynthMT and **(b)** unseen, real data. SAM-3Text+HPO preserves both low and high values across the full range of lengths and curvatures, as reflected by low KL divergence values computed from these histograms. Distributions for the other methods on SynthMT are shown in Fig E in S1 Appendix and Fig F in S1 Appendix.

with the findings of Pachitariu et al. [15]), they easily beat generalist models such as SAM or SAM2 on these biologically relevant tasks. As expected, StarDist, although one of the fastest with ≈ 5 images per second, performs the worst with its default settings, since it is architecturally made for star-convex shapes. However, after tuning its hyperparameters it still outperforms FIESTA.

### Tests on unseen, real IRM data with human baseline further confirm SAM3Text

Analyzing performances on unseen, real data and comparing them to a human baseline allows us to put the previously presented numbers on the synthetic SynthMT dataset into context.

Fig 7 demonstrates that SAM3, provided with a simple text prompt (*"thin line"*) and with optimized hyperparameters using 10 images from our synthetic dataset SynthMT, maintains strong performance on unseen, real images and achieves segmentation quality comparable to human performance. It reaches a mean SKIoU of $0.74 \pm 0.18$, compared to a human of $0.8 \pm 0.14$, placing the model's performance within the variability observed between human annotators. Although other methods also benefit from HPO, none reaches this level of performance. The methods that do not improve through HPO are the same as those found above in the analysis on SynthMT (TARDIS, CellSAM, SAM). Cellpose-SAM is the only method that improves on synthetic data, but worsens on unseen, real data.

Table 2 confirms these findings through the quantitative biological metrics, where SAM3Text consistently improves with HPO. It matches and slightly surpasses human inter-annotator agreement in counting accuracy, with a deviation of only 1.17 instances per image relative to the Ground Truth (GT), and achieves sub-micrometer accuracy in length estimation (deviation of $0.3\,\mu m$). For curvature, SAM3Text remains within the inter-annotator range, achieving a better average ($0.29\,\mu m^{-1}$ deviation) and only a marginally higher KL divergence.

However, all models adapted for microscopy — except StarDist — achieve reasonable downstream performance. In particular, CellSAM already performs strongly in its default configuration, ranking second in SKIoU. It also shows good agreement with the ground truth for length and curvature, but underperforms in counting accuracy. In contrast, only SAM3Text consistently attains top or near-top performance across all biological metrics without such compromises.

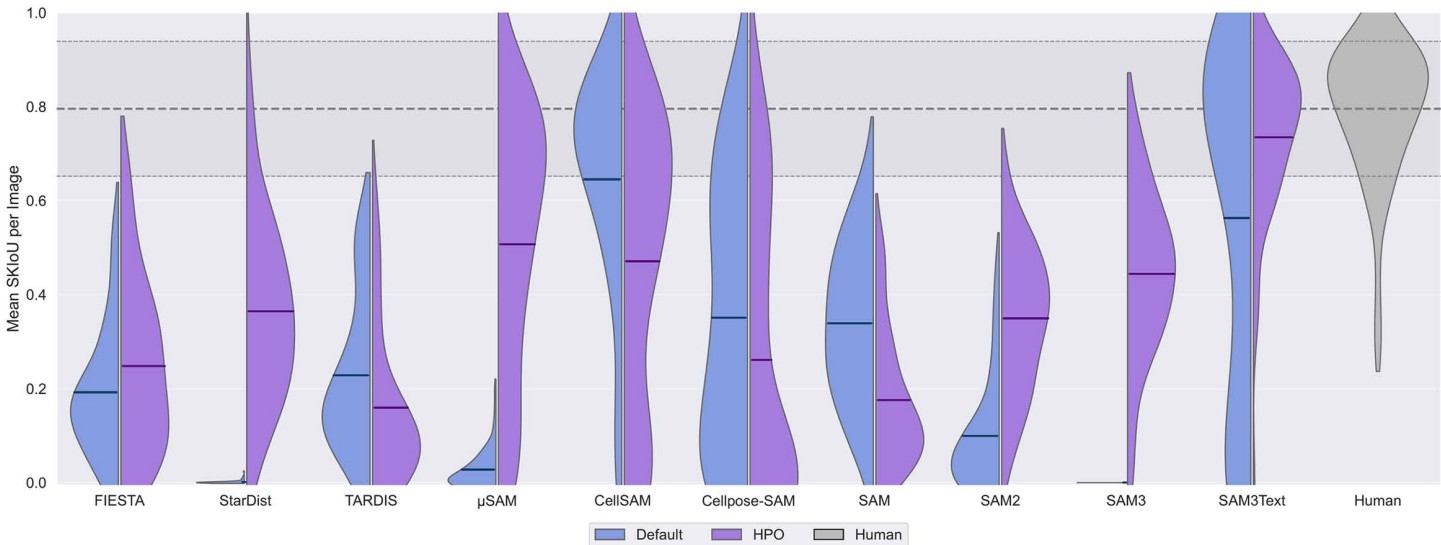

**Fig 7. Only SAM3Text + HPO reaches human segmentation performance on unseen, real IRM data.** Split violin plots show the distribution of per-image SKIoU scores ($n = 66$) for each method evaluated on unseen, real IRM data. The left side of each violin (blue) represents the default configurations, while the right side (purple) shows the performance after HPO on 10 random, synthetic synthetic images from `SynthMT`. Each violin includes a horizontal line indicating the mean SKIoU across all images. The human performance is shown as a solid gray violin for reference. Its mean and standard deviation values are indicated by horizontal lines across the plot. The plot shows that the optimized SAM3Text matches the human inter-annotator baseline. While other methods also improve with HPO, none demonstrates the top-tier performance of SAM3Text. Notably, CellSAM already approaches human-level performance in its default configuration, but exhibits decreased performance after HPO.

## Qualitative analysis reinforces the strength of SAM3Text

We examine a particularly challenging real example with many intersecting MTs and low SNR in Fig 8 to qualitatively compare methods, and find that these observations are in line with the quantitative results reported before.

Among all methods tuned with HPO on synthetic data, SAM3Text clearly stands out. It consistently captures all visible MTs (see bottom right in Fig 8), including most of the complex intersections, producing continuous and well-separated instances with only very few false positive or false negative detections.

Its predecessors exhibit diverse failure modes: SAM generates numerous small, spurious detections despite post-processing, while SAM2 merges intersecting filaments and occasionally segments enclosed areas, producing incorrect, non-elongated shapes.

The anchor-point-based methods FIESTA and TARDIS, which were explicitly designed for MT analysis, show mixed performance. Both methods can resolve intersections, but they miss a substantial number of filaments. Moreover, when instances are detected, endpoints are often not identified correctly, leading to inaccurate length estimates.

Among the other SAM-based methods specialized for microscopy, $\mu$SAM, CellSAM, and Cellpose-SAM recover many filament structures but exhibit notable weaknesses. $\mu$SAM and CellSAM frequently merge intersecting MTs, resulting in under-segmentation that hinders downstream measurements. CellSAM additionally produces overly thick segmentations. It is worth recalling that our SKIoU metric is robust to such width inaccuracies by design, as it operates on skeletonized versions of the predictions. Cellpose-SAM performs best within this group, correctly identifying a large fraction of instances, but still misses a large number of filaments, which limits its suitability for fully automated analysis pipelines that require high instance recall. Finally, StarDist fails to capture the elongated morphology of MTs as expected, since its star-convex object prior biases predictions toward circular shapes.

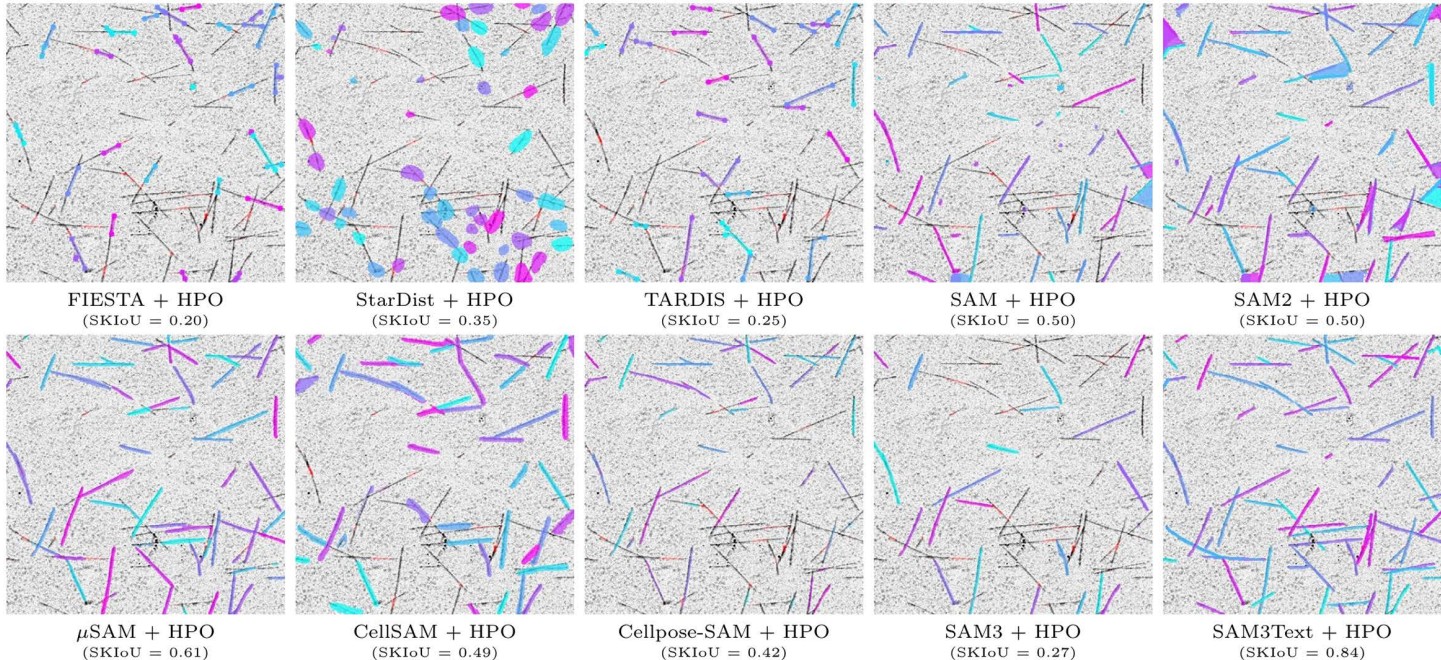

**Fig 8. Qualitative comparison on an unseen, real-world *in vitro* reconstituted MT assay.** For each method, we show predictions after HPO on 10 synthetic images from `SynthMT` (few-shot setting). The selected real image is particularly challenging, as it contains many intersecting MTs and exhibits a low signal-to-noise ratio (SNR), exposing a wide range of failure modes. For anchor-point methods such as FIESTA and TARDIS, only the first and last predicted points per instance are shown for visual clarity. Underneath each image we report the mean SKIoU value for this specific image, in order to correlate it with a visual impression. SAM3Text clearly performs best in this setting, while all other methods show limitations that may hinder their suitability for large-scale fully automated analysis. For more comparisons and dynamic exploration of this kind, we refer to our project page at DATEXIS.github. io/SynthMT-project-page.

## Discussion

### Suitability of automated microtubule (MT) analysis

A central question in MT image analysis is whether fully automated, expert-level segmentation is currently achievable. Our results provide a nuanced, previously unknown answer. Most segmentation approaches still fall short of the accuracy required for reliable downstream biological measurements on interference reflection microscopy (IRM) data of *in vitro* reconstituted MTs. They frequently fragment filaments, miss faint or short instances completely, or introduce spurious detections, even when Hyperparameter Optimization (HPO) is used (few-shot setting).

### SAM3 altered the picture

However, our study highlights a key positive outcome: guiding the recently introduced SAM3 model by a simple text prompt ("thin line") and tuning it on only 10 synthetic `SynthMT` images reaches performance within the human inter-annotator range on unseen, real data. This is, to our knowledge, the first demonstration that a general-purpose vision foundation model can be adapted to deliver fully automated, biologically meaningful MT segmentation.

### Tuning with `SynthMT` enables human-grade performance

A key enabler of this achievement is the synthetic dataset `SynthMT`, which can be generated entirely from real MT images without the need for manual annotations. It automatically captures the visual statistics of real IRM frames and uses these

to simulate filamentous structures. This enables the adaptation of segmentation methods to new microscopes, imaging conditions, or laboratories without any annotation effort. The success of SAM3Text after HPO on only 10 of those images demonstrates that method configuration can be effectively guided toward human-grade performance, without requiring any manual annotation of real data. This provides direct evidence of synthetic-to-real transfer: hyperparameters optimized exclusively on synthetic images generalize to unseen, real IRM images.

Within this context, `SynthMT` serves as both a benchmark and a diagnostic tool. It exposes the typical failure modes of existing methods, including fragmentation, missed instances, and sensitivity to domain shifts, and it provides a controlled setting for evaluating and tuning segmentation methods.

## Summary

`SynthMT` provides a reproducible and annotation-free framework for evaluating segmentation methods, identifying their limitations, and enabling powerful foundation models like SAM3 to be configured for filamentous microscopy data. This establishes a practical and fully automated route toward scalable MT segmentation suitable for high-throughput biological experiments, such as (non-dynamic) end-point experiments.

## Limitations and future work

One limitation is that our evaluation focuses exclusively on fully automated segmentation methods. Semi-automated approaches that require human input or corrections are not considered.

Moreover, our synthetic image generation relies on a parametric approach derived from real MT images. While it avoids the need for manual annotation, alternative approaches such as diffusion-based models have shown potential for generating realistic microscopy images [32], but they typically require large annotated datasets for training. Investigating these methods as a complementary or alternative generation strategy is an avenue for future work.

Furthermore, we only use `SynthMT` for HPO-based few-shot adaptation in this work, not as a large-scale training dataset for full model fine-tuning. Fully supervised training on `SynthMT` is an open direction that could further improve model performance. To facilitate this, both `SynthMT` and our human-annotated, real IRM dataset are made publicly available at Hugging Face.

Lastly, `SynthMT` currently focuses on 2D image generation from single static frames, which does not capture the temporal dynamics of MTs such as their alternating phases of growth and shrinkage. Extending the pipeline to videos would allow instance tracking over time, which is crucial for studying MT kinematics. Several approaches could be explored to achieve robust MT tracking. Time could be treated as a third dimension in volumetric models such as AnyStar [20] or Cellpose-SAM [15], or fully automated segmentation outputs could be combined with existing tracking algorithms [6,9,67]. Notably, SAM2 and SAM3 already support video data, which opens the possibility of extending their promptable, fully automated segmentation capabilities to temporal MT datasets.

## Supporting information

**S1 Appendix. Full supplementary information including method details, preprocessing parameters, experimental protocol, example images, perceptual study interface, length and curvature distributions, ablation studies, and hyperparameter optimization results. Fig A. Examples from the `SynthMT` dataset.** 18 images sampled from the synthetic `SynthMT` dataset. Each image depicts individual MTs growing from stabilized seeds (shown in red) under simulated IRM conditions. The dataset captures natural variation in MT quantity, length, and curvature across different experimental conditions represented by 660 optimized parameter sets. Every image is accompanied by pixel-accurate segmentation masks for each MT, providing ground-truth annotations for quantitative benchmarking of segmentation methods. **Fig B. Representative samples from related synthetic MT datasets.** (a) MicSim_FluoMT: Six $666 \times 666$ examples from fluorescence microscopy simulations of *in vivo* astral MTs during *C. elegans* mitosis. (b) DRIFT: Six $512 \times 512$ synthetic

images generated for generic curved filament segmentation. Both datasets differ fundamentally from ours in imaging modality, biological context, structural complexity, and scope. **Fig C. Web-based interface for expert assessment of perceptual realism of synthetic microscopy images.** Participants viewed a single image at a time and rated it along five predefined dimensions using a 7-point Likert scale. Images were presented in randomized order, and each image was rated exactly once per participant. **Fig D. Representative images from the human validation study.** Nine examples from the evaluation set shown to domain experts: three real IRM images (top row), three synthetic images from `Syn-thMT` (middle row), and three synthetic images from DRIFT (bottom row). **Fig E.** Predicted and ground-truth distributions of length and curvature for all methods with their default parameters. **Fig F.** Predicted and ground-truth distributions of length and curvature for all methods with tuned parameters obtained through HPO. **Fig G. HPO trajectories highlight rapid optimization.** Best Skeleton Intersection over Union (SKIoU) value observed up to each of the 1000 HPO trials for each method, using 10 images from `SynthMT` for optimization. **Fig H. A few key hyperparameters drive method performance.** Parameter importance for all models, calculated via f-ANOVA on the 10 `SynthMT` images used for optimization. **Table A.** Optimized FIESTA hyperparameters obtained through HPO. **Table B.** Default StarDist hyperparameters and those obtained through HPO. **Table C.** Default TARDIS hyperparameters and those obtained through HPO. **Table D.** Default SAM hyperparameters and those obtained through HPO. **Table E.** Default SAM2 hyperparameters and those obtained through HPO. **Table F.** Default $\mu$SAM hyperparameters and those obtained through HPO. **Table G.** Default CellSAM hyperparameter and that obtained through HPO. **Table H.** Default Cellpose-SAM hyperparameters and those obtained through HPO. **Table I.** Default SAM3 hyperparameters and those obtained through HPO. **Table J.** Default SAM-3Text hyperparameters and those obtained through HPO. **Table K.** Default and tuned preprocessing parameters obtained through HPO for all methods. **Table L.** Comparison of HPOs for SAM3Text using varying numbers $N$ of synthetic images. Performance saturates rapidly beyond $N$ = 10. **Table M.** Comparison of HPOs for SAM3Text using three different data sources: `SynthMT` images, binary DRIFT samples, and human-annotated real images from the test set.
(PDF)

## Acknowledgments

We would like to thank Dominik Fachet and Gil Henkin from the Reber lab for providing data, and also thank the further study participants Moritz Becker, Nathaniel Boateng, and Miguel Aguilar. The Reber lab thanks staff at the Advanced Medical Bioimaging Core Facility (Charité, Berlin) for imaging support. Furthermore, we thank Kristian Hildebrand and Chaitanya A. Athale (IISER Pune, India) and his lab for helpful discussions.

## Author contributions

**Conceptualization:** Mario Koddenbrock, Justus Westerhoff, Simone Reber, Felix A Gers, Erik Rodner.

**Data curation:** Mario Koddenbrock, Justus Westerhoff, Dominik Fachet.

**Formal analysis:** Mario Koddenbrock, Justus Westerhoff.

**Funding acquisition:** Simone Reber, Felix A Gers, Erik Rodner.

**Investigation:** Mario Koddenbrock, Justus Westerhoff.

**Methodology:** Mario Koddenbrock, Justus Westerhoff.

**Project administration:** Mario Koddenbrock, Justus Westerhoff.

**Resources:** Mario Koddenbrock, Justus Westerhoff, Dominik Fachet.

**Software:** Mario Koddenbrock, Justus Westerhoff.

**Supervision:** Simone Reber, Felix A Gers, Erik Rodner.

**Validation:** Mario Koddenbrock, Justus Westerhoff.

**Visualization:** Mario Koddenbrock, Justus Westerhoff.

**Writing – original draft:** Mario Koddenbrock, Justus Westerhoff, Dominik Fachet.

**Writing – review & editing:** Mario Koddenbrock, Justus Westerhoff, Simone Reber, Felix A Gers, Erik Rodner.

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
