## [Decision Letter · Decision Letter 0]

12 Feb 2026

PCOMPBIOL-D-26-00044

Synthetic data enables human-grade microtubule analysis with foundation models for segmentation

PLOS Computational Biology

Dear Dr. Koddenbrock,

Thank you for submitting your manuscript to PLOS Computational Biology. After careful consideration, we feel that it has merit but does not fully meet PLOS Computational Biology's publication criteria as it currently stands.

The reviewers appreciated the method for generating synthetic MT images and the evaluation of existing segmentation methods. However, in addition to detailed comments, all three reviewers expressed concern about the lack of comparison with real data. This raises the question of whether the synthetic images are representative of real microtubule images, which appears to be a significant limitation of the current version of the manuscript

Therefore, we invite you to submit a revised version of the manuscript that addresses the points raised during the review process.

We look forward to receiving your revised manuscript.

Kind regards,

Dimitrios Vavylonis

Section Editor

PLOS Computational Biology

**Journal Requirements:**

3) We notice that your supplementary Figures, and Tables are included in the manuscript file. Please remove them and upload them with the file type 'Supporting Information'. Please ensure that each Supporting Information file has a legend listed in the manuscript after the references list.

**Reviewers' comments:**

Reviewer's Responses to Questions

**Comments to the Authors:**

Reviewer #1: The paper introduces SynthMT, a realistic synthetic dataset for microtubule segmentation, enabling systematic benchmarking without manual annotations. Most existing methods evaluated by the author performed poorly, but SAM3 with text prompting, tuned on only a few synthetic images, reaches human-level accuracy on real data, showing that fully automated microtubule segmentation is now feasible using well-designed synthetic data.

Strengths:

- I would like to highlight (and this is not common) that all resources to this paper were released, as well as reader is supported with an interactive website that allow for easy and fast exploration of the research. Something that in my humble opinion should nowadays be obligatory.

Need corrections in reviewer opinion:

- The term “foundation model” is used too broadly in the paper, and it is not strictly correct for several of the evaluated methods.

• SAM / SAM2 / SAM3 clearly qualify as foundation models.

• CellSAM, µSAM, Cellpose-SAM are task-specific adaptations or fine-tuned derivatives of SAM for microscopy, rather than foundation models themselves.

• TARDIS is a domain-specific model trained for microtubules and does not meet the criteria of a foundation model.

- L15-16 - Is there any citation that can be used here to justified this claim?

-L73-L80 - Could be good to mention also a physical simulation of microscopy images () or GassianSplating?

-L135 - This could be clarified as 42% is true for 3D model benchmark on Cryo-EM/EM datastes not TIRF.

-L203 - The authors mention that image crops containing a significant number of imaging artifacts were manually discarded. While this is reasonable for defining a clean target distribution, it would be helpful if the authors could elaborate on whether this choice was evaluated empirically. In particular, removing artifact-heavy images may improve performance on “ideal” samples, but it could also reduce robustness at inference time, as real experimental data often contain such artifacts. Did the authors test whether excluding these crops affects generalization to lower-quality or artifact-contaminated images, or consider including a controlled fraction of artifacts in the synthetic data generation? A short discussion of this trade-off, even if not experimentally explored, would strengthen the methodological justification.

-L289 - Here other I do not understand it and it should be clarified but, as i understand author tries to use TARDIS to predict 2D images and output 2D semantic mask (binary). This is possible with the code:

"""

tardis_mt_tirf -dir <path-to-your-images> -out tif_None

"""

But I think what author tries to say is that TARDIS does not allow to output array mask of instances (not binary), alought this should be possible passing -out None_tif.

- SKIoU -The use of SKIoU is well motivated and appropriate for thin filament segmentation. One remaining concern is that SKIoU may still penalize predictions where the skeleton centerline is consistently but slightly shifted relative to the ground truth (e.g. by a few pixels), even if the filament length and topology are otherwise correct. Could the authors comment on whether this effect was observed in practice? It may be worth discussing whether introducing a small spatial tolerance (e.g. allowing a few-pixel offset when computing skeleton overlap) was considered, and why the strict formulation of SKIoU was ultimately chosen.

- Since the authors explicitly encourage the community to use SynthMT beyond benchmarking, it would strengthen the paper to include at least a limited experiment demonstrating that the synthetic data can improve model performance when used for training. For example, the authors could train a lightweight CNN-based segmentation model or retrain TARDIS from scratch using (i) only real data, (ii) only SynthMT, and (iii) a combination of real and synthetic data, and report basic metrics such as F1, precision, recall, or AUC on a held-out real test set. Even a small-scale experiment would provide concrete evidence that SynthMT is not only a benchmarking resource but also beneficial for model development and domain adaptation.</path-to-your-images>

Reviewer #2: I thank the authors for their clear and well written paper. But, I am concerned about the novelty of this work as the new addition is the generation of synthetic MT images. Existing methods are tested against each other, but I am not sure this would warrant publication by itself, though the results as to which existing methods perform better than others is interesting. The novelty of the paper seems to largely depend on whether the synthetic images are 1) accurate to real systems and 2) useful enough to make it worth it to generate them. They generally seem accurate (though I have some concerns as listed below), but I am more concerned about the usefulness of the SynthMT method. More detailed comments are listed below.

• If HPO with only 10 synthetic images is enough to make the existing methods trace MTs more accurately this seems to somewhat undermine the usefulness of SynthMT. Aren’t there at least 10 labeled, real MT images available that could be used in place of the SynthMT images to get the same result? Does doing HPO on more images improve the methods further or does the accuracy begin to saturate at only a few images?

• Linear segments should be among the easiest aspects in an image to identify, would generating simple images with lines and feeding them into the various methods to do the HPO also work to enhance the method performance for labeling real MT images? If so there seems to be limited usefulness of SynthMT unless I am missing something.

• Fluorescent MT images are often in a single channel, yet the synthetic images seem to have color in them. Is there a reason why color added to these images and would this affect the ability of the algorithm to detect linear segments?

• Is red the fixed MT seed even in Fig. A.7? Some of them appear much longer than other, is this just by chance?

Reviewer #3: In the manuscript "Synthetic data enables human-grade microtubule analysis

with foundation models for segmentation", the authors describe a series of experiments to support the segmentation of filamentous structures (lines, strokes) in 2D images that contain nothing but such structures and noise with existing foundation models.

Contributions:

1. They built an ad-hoc generator for such images that resembles the appearance of microtubules (MT) imaged with IRM.

2. They used the generator to generate the dataset SynthMT that can be used to train and test deep learning based methods to automatically label such images.

3. They use this dataset to evaluate the performance of nine fully automatic methods to segment the MTs, notably without using the dataset to train any of those methods. They improve the performance of most methods using HPO (method and pre-/ post-processing parameters) using 1000 trials on 10 images. They find that only SAM3 with text prompts and HPO works well enough to be practically useful (CellSAM **without** HPO gets pretty close though)

They made their dataset, image generator, and evaluation code publicly available as open source/ data under permissive licenses.

This is a technically sound performance study how well currently available segmentation foundation models can identify microtubules in 2D IRM images. The most important results are that SAM3 models with an appropriate text prompt are getting very closer to human performance than the other tested models.

The authors created an ad-hoc image generator that they tuned to create images that somehow resemble the target images, and used this to create a benchmark dataset. They tune hyperparameters of the assessed methods with a few examples of this benchmark dataset.

The paper is easy to read, clear, and comprehensive in details.

The major missing piece, if I didn't miss anything, is that I did not see the performance comparison on real data (Tab 2) **without** HPO. HPO on SynthMT improving this performance would be the ultimate argument in support of the utility of the benchmark dataset. Without this comparison, the paper shows that:

1. foundation models are good at segmenting microtubules in their artificial images and also in real images

2. HPO on representative data sometimes makes things better (shown in their artificial images)

Particularly 2 is a little weak and sort of expected. One could add

3. fine-tuning models on representative data improves performance

which is not the goal of this paper, but it's similarly expected. One could argue that, without this comparison, the benchmark dataset in its current form is irrelevant.

Can you please add this comparison and/ or describe and highlight it more prominently?

Minor: relating to existing benchmark generation methods could be informative where this method sits (specifically CARE and microsim come to mind).

Showing that DRIFT images look less similar to microscopy images thant the noisier SynthMT images is a bit obvious and could be discussed in a less prominent place.

Notes:

The manuscript is motivated by automatcially reconstructing microtubules in 2D microscopy images. Lack of a sufficiently large dataset for training or validation is identified as a major bottleneck to address this problem.

Related work section is comprehensive and enumerates relevant tools to segment microtubules and existing benchmark datasets for similar 2D images.

Image generator consists of an ad-hoc polyline segment generator, spot generator, noise generator and an image distortion module. The low dimensional parameter vector \theta of this generator is tuned to minimize the Cosine distance between the DINOv2 embeddings of real microscopy images and generated images. Since the generation method is not differentiable, gradient descent cannot be used and so they use the tree structured Parzen estimator to iteratively wiggle the parameters into a local optimum. Domain experts checked the results and found them more convincing than DRIFT images which are binary line drawings, e.g. an easy bar to pass.

Similar to existing microscopy simulators as used for e.g. CARE, or [microsim](https://talleylambert.com/microsim/), the described method uses an ad-hoc hard-coded model for the expected structures (sort of stiff lines), but other than thos methods does not use a physically plausible model to generate realistic images but a simple render, add noise, warp process.

The compared methods are the SAM based SAM, SAM2, microSAM, CellSAM, Cellose-SAM, SAM3, TARDIS, and the baseline classical method FIESTA, a line detector.

All SAM methods were used without prompts except for SAM3 which was told to detect "thin lines". FIESTA was set to look for 3px wide lines.

Naive evaluation metrixcs are SKIoU, AP, F1 generated from pixel masks. "Biological" metrics are number of detections, length, curvature, i.e. meaningful analysis targets, compared with mean and STD and also KL divergence. Combining this into a single score in Tab2 (geom mean?) could help to get a clearer picture which method performs the best.

Compute efficiency is reported on an A100 PC, except for FIESTA which was run on a Macbook Pro M3.

**Have the authors made all data and (if applicable) computational code underlying the findings in their manuscript fully available?**

Reviewer #1: Yes

Reviewer #2: Yes

Reviewer #3: Yes

PLOS authors have the option to publish the peer review history of their article (what does this mean?). If published, this will include your full peer review and any attached files.

Reviewer #1: **Yes:** Dr. rer. Medic. Robert Kiewisz

Reviewer #2: No

Reviewer #3: No

**Figure resubmission:**
---

## [Decision Letter · Decision Letter 1]

7 Apr 2026

Dear Mr. Koddenbrock,

We are pleased to inform you that your manuscript 'Synthetic data enables human-grade microtubule analysis with foundation models for segmentation' has been provisionally accepted for publication in PLOS Computational Biology.

Some of the references did not compile properly and appear as "??". Please check the manuscript thoroughly for similar errors.

Best regards,

Dimitrios Vavylonis

Section Editor

PLOS Computational Biology

Reviewer's Responses to Questions

**Comments to the Authors:**

Reviewer #1: No further comment. I would like to thank authors for responding to all my comments.

Reviewer #2: I thank the authors for addressing my comments completely. I have no further comments.

**Have the authors made all data and (if applicable) computational code underlying the findings in their manuscript fully available?**

Reviewer #1: Yes

Reviewer #2: Yes

PLOS authors have the option to publish the peer review history of their article (what does this mean?). If published, this will include your full peer review and any attached files.

Reviewer #1: **Yes:** Dr. rer. Medic. Robert Kiewisz

Reviewer #2: No

---

## [Editor Report · Acceptance letter]

PCOMPBIOL-D-26-00044R1

Synthetic data enables human-grade microtubule analysis with foundation models for segmentation

Dear Dr Koddenbrock,

I am pleased to inform you that your manuscript has been formally accepted for publication in PLOS Computational Biology. Your manuscript is now with our production department and you will be notified of the publication date in due course.

With kind regards,

Sharmila Kamatchi
